# *Salmonella* exploits host- and bacterial-derived β-alanine for replication inside host macrophages

**Shuai Ma[†], Bin Yang[†], Yuyang Sun, Xinyue Wang, Houliang Guo, Ruiying Liu, Ting Ye, Chenbo Kang, Jingnan Chen, Lingyan Jiang***

National Key Laboratory of Intelligent Tracking and Forecasting for Infectious Diseases, TEDA Institute of Biological Sciences and Biotechnology, Nankai University, Tianjin, China

## eLife Assessment

The authors use a multidisciplinary approach to provide a link between Beta-alanine and S. Typhimurium (STM) infection and virulence. This **valuable** work shows how Beta-alanine synthesis mediates zinc homeostasis regulation, possibly contributing to virulence. The work is **convincing** as it adds to the existing knowledge of metabolic flexibility displayed by STM during infection.

**\*For correspondence:**
jianglingyan@nankai.edu.cn

[†]These authors contributed equally to this work

**Competing interest:** The authors declare that no competing interests exist.

**Abstract** *Salmonella* is a major foodborne pathogen that can effectively replicate inside host macrophages to establish life-threatening systemic infections. *Salmonella* must utilize diverse nutrients for growth in nutrient-poor macrophages, but which nutrients are required for intracellular *Salmonella* growth is largely unknown. Here, we found that either acquisition from the host or de novo synthesis of a nonprotein amino acid, β-alanine, is critical for *Salmonella* replication inside macrophages. The concentration of β-alanine is decreased in *Salmonella*-infected macrophages, while the addition of exogenous β-alanine enhances *Salmonella* replication in macrophages, suggesting that *Salmonella* can uptake host-derived β-alanine for intracellular growth. Moreover, the expression of *panD*, the rate-limiting gene required for β-alanine synthesis in *Salmonella*, is upregulated when *Salmonella* enters macrophages. Mutation of *panD* impaired *Salmonella* replication in macrophages and colonization in the mouse liver and spleen, indicating that de novo synthesis of β-alanine is essential for intracellular *Salmonella* growth and systemic infection. Additionally, we revealed that β-alanine influences *Salmonella* intracellular replication and in vivo virulence partially by increasing expression of the zinc transporter genes *znuABC*, which in turn facilitates the uptake of the essential micronutrient zinc by *Salmonella*. Taken together, these findings highlight the important role of β-alanine in the intracellular replication and virulence of *Salmonella*, and *panD* is a promising target for controlling systemic *Salmonella* infection.

## Introduction

*Salmonella* is a major foodborne pathogen worldwide that can cause self-limiting gastroenteritis or life-threatening systemic disease in a wide range of animals (*Fàbrega and Vila, 2013*; *Ohl and Miller, 2001*). *Salmonella* infection remains a significant global public health concern. An estimated 93.8 million cases of gastroenteritis and 27 million cases of systemic diseases caused by *Salmonella* species occur annually worldwide, with 355,000 deaths (*Kim et al., 2019*; *Majowicz et al., 2010*). The ability to survive and replicate in host macrophages is a key determinant for *Salmonella* to induce systemic infection (*Fields et al., 1986*; *LaRock et al., 2015*; *Leung and Finlay, 1991*).

**eLife digest** *Salmonella*, a type of bacterium, is one of the most common foodborne pathogens and is responsible for illnesses ranging from gastroenteritis to typhoid fever. Each year, it infects over 100 million people globally, leading to 350,000 deaths.

Immune cells called macrophages are important for defending against bacterial infections as they can engulf and destroy harmful bacteria. However, *Salmonella* is able to survive and multiply within these very immune cells that are meant to eliminate it. To do so, the bacteria require nutrients such as amino acids, which are the building blocks of proteins. These are either produced by the bacteria or obtained from the infected host. However, the specific nutrients that *Salmonella* require to survive and multiply, as well as their source, remained unknown.

To investigate, Ma, Yang et al. measured amino acid levels in macrophages that had been infected with *Salmonella* and compared them to those in uninfected macrophages. This revealed that the levels of an amino acid called β-alanine – which differs from many amino acids because it is not used to make proteins – are lower in infected macrophages. Furthermore, providing infected macrophages with more β-alanine increased bacterial replication. This suggests that the bacteria acquire this amino acid from the macrophages in order to survive and replicate.

To determine whether *Salmonella* can also make β-alanine themselves, Ma et al. prevented them from producing it, which slowed bacterial growth and led to milder infections in mice. This ability to produce β-alanine required a gene known as *PanD*. Further experiments also showed that β-alanine assists *Salmonella* in acquiring the essential micronutrient zinc from macrophages.

Taken together, the findings of Ma et al. reveal the critical role of β-alanine in *Salmonella* growth within macrophages and its ability to cause disease. The *panD* gene that enables *Salmonella* to synthesize β-alanine could serve as a potential target for new treatments or vaccines. By targeting this specific aspect of *Salmonella's* survival strategy, researchers may be able to develop more effective methods to prevent and treat these dangerous infections.

After internalization by macrophages, *Salmonella* delivers a set of more than 30 effector proteins to the macrophage cytoplasm, mainly through a type III secretion system (T3SS) encoded by *Salmonella* pathogenicity island-2 (SPI-2) (*Pillay et al., 2023*). SPI-2 effectors manipulate diverse cellular processes to promote the formation of a membrane-bound compartment, termed the *Salmonella*-containing vacuole (SCV), a niche where *Salmonella* resides and grows (*Castanheira and García-del Portillo, 2017*; *Rosenberg et al., 2022*; *Steeb et al., 2013*). SCV protects *Salmonella* from contact with antimicrobial agents in macrophages (*Figueira and Holden, 2012*; *Li et al., 2023*). Moreover, SPI-2 effectors induce the formation of specific tubular membrane compartments that extend from the SCV, known as *Salmonella*-induced filaments (SIFs). These filaments allow *Salmonella* to access various types of endocytosed nutrients, thereby facilitating efficient replication within macrophages (*Brumell et al., 2001*; *Liss et al., 2017*; *Rajashekar et al., 2008*).

As the SCV of macrophages is a nutrient-poor environment (*Fields et al., 1986*; *Kehl et al., 2020*), to effectively replicate in the SCV, *Salmonella* needs to acquire a wide range of host nutrients or host-derived metabolites and synthesize metabolites de novo that cannot be sufficiently accessed from the host (*Dandekar et al., 2014*; *Röder et al., 2021*; *Tuli and Sharma, 2019*). Nutrients/metabolites are used by intracellular *Salmonella* either as carbon sources to generate energy or for the synthesis of fatty acids and proteins (*Dandekar et al., 2014*; *Steeb et al., 2013*). Moreover, several metabolites were found to be employed by *Salmonella* as environmental cues to induce the expression of virulence genes (*Jiang et al., 2021*; *Wang et al., 2023*). In recent years, an increasing number of studies have focused on the intracellular nutrition of *Salmonella* (*Bumann and Schothorst, 2017*; *Liss et al., 2017*; *Röder et al., 2021*); however, the nutrients that are required for *Salmonella* replication in macrophages remain largely unknown.

β-Alanine, also known as 3-aminopropionic acid (3-AP), is the only naturally occurring β-type amino acid and is found in all living organisms (*Song et al., 2023*). β-Alanine can be synthesized de novo by bacteria, fungi, and plants, whereas animals need to obtain it from food or generate it via the catabolism of cytosine and uracil (*Gojković et al., 2001*; *Wang et al., 2021b*). In bacteria, β-alanine is synthesized *via* the decarboxylation of L-aspartate, a reaction catalyzed by L-aspartate decarboxylase (PanD)

(*Begley et al., 2001*; *Schmitzberger et al., 2003*; *West et al., 1985*). The *panD* gene is conserved among most bacteria (*Nozaki et al., 2012*). Although β-alanine is a nonprotein amino acid that is not incorporated into proteins, it has important physiological functions in the metabolism of organisms (*Wang et al., 2021b*; *Yuan et al., 2022*). First, β-alanine forms a part of pantothenate (vitamin B5), which is the key precursor for the biosynthesis of coenzyme A (CoA) (*Webb et al., 2004*; *White et al., 2001*). CoA is an essential cofactor involved in many metabolic pathways, including the synthesis and degradation of fatty acids, pyruvate oxidation through the tricarboxylic acid (TCA) cycle, and the production of secondary metabolites (*Davaapil et al., 2014*; *Gout, 2019*; *Sibon and Strauss, 2016*; *Theodoulou et al., 2014*). Second, β-alanine is a limiting precursor of carnosine, a nonenzymatic free radical scavenger and a natural antioxidant, with anti-inflammatory and neuroprotective effects in animals (*Boldyrev et al., 2013*; *Hoffman et al., 2018*). In the past 15 y, β-alanine has become one of the most commonly used sports supplements worldwide (*Bellinger, 2014*; *Hoffman et al., 2018*; Huerta Ojeda, Tapia *Huerta Ojeda et al., 2020*). Although both *Salmonella* and host cells are capable of producing β-alanine, whether β-alanine contributes to the pathogenicity and intracellular growth of *Salmonella* remains unknown.

In this work, using targeted metabolic profiling, in vitro and in vivo infection assays, and many other molecular techniques, we demonstrated that the utilization of β-alanine is essential for *Salmonella* replication in host macrophages and virulence in mice. *Salmonella* acquires β-alanine both via the uptake of β-alanine from host macrophages and the de novo synthesis of β-alanine. Further investigation revealed the molecular mechanism underlying the contribution of β-alanine to *Salmonella* intracellular replication and pathogenicity, wherein β-alanine promotes the expression of zinc transporter genes to facilitate the uptake of the essential micronutrient zinc by intracellular *Salmonella*, therefore promoting *Salmonella* replication in macrophages and subsequent systemic infection. Taken together, these findings demonstrate a correlation between *Salmonella* β-alanine utilization and zinc uptake during intracellular infection and provide new insights into the intracellular nutrition of *Salmonella*. The rate-limiting gene (*panD*) in the β-alanine synthesis pathway of *Salmonella* might be a future target for the prevention and treatment of this pathogen.

## Results

### Host-derived β-alanine promotes *Salmonella* replication inside macrophages

To explore changes in the levels of different amino acids inside macrophages upon *Salmonella* infection, we performed targeted metabolomics analysis of mouse RAW264.7 macrophages that were mock-infected or infected with wild-type *Salmonella* (*Salmonella enterica* serovar Typhimurium ATCC 14,028 s, STM) for 8 hr using liquid chromatography-tandem mass spectrometry (LC–MS/MS) (*Figure 1A*). Principal component analysis (PCA) demonstrated a clear separation between the mock- and *Salmonella*-infected groups (*Figure 1B*). A total of 26 free amino acids were analyzed, and eight showed significant differences in abundance between the two groups (VIP (Variable Importance in the Projection) >1 and p<0.05; FC (fold change) > 1.5 or < 0.667) (*Figure 1C and D*). Compared with those in the mock-infected group, the concentrations of three amino acids (L-hydroxyproline, L-citrulline and L-cysteine) were upregulated (*Figure 1C*), and five amino acids (L-asparagine, L-serine, L-aspartate, β-alanine and γ-aminobutyric acid) were downregulated (*Figure 1D*) in the *Salmonella*-infected group. Consistent with previous findings, intracellular serine concentrations were downregulated due to the reprogramming of macrophage glucose metabolism during *Salmonella* infection (*Jiang et al., 2021*). *Salmonella* can use host-derived aspartate and asparagine for growth in macrophages (*Popp et al., 2015*); therefore, the decrease in intracellular aspartate and asparagine upon *Salmonella* infection is likely due to their utilization by bacteria. Interestingly, β-alanine concentrations were also downregulated in the *Salmonella*-infected group (*Figure 1D*), suggesting that intracellular *Salmonella* may use host-derived β-alanine for growth.

To investigate whether host-derived β-alanine can promote intracellular *Salmonella* replication, we added an additional 0.5, 1, 2, 4 mM β-alanine (*Schneider et al., 2004*) to the culture medium (RPMI) of RAW264.7 cells and then infected them with *Salmonella* to test the influence of β-alanine addition on the ability of *Salmonella* to replicate in macrophages. The results showed that the replication of *Salmonella* in RAW264.7 cells significantly (p<0.001) increased with the addition of 1, 2, or 4 mM

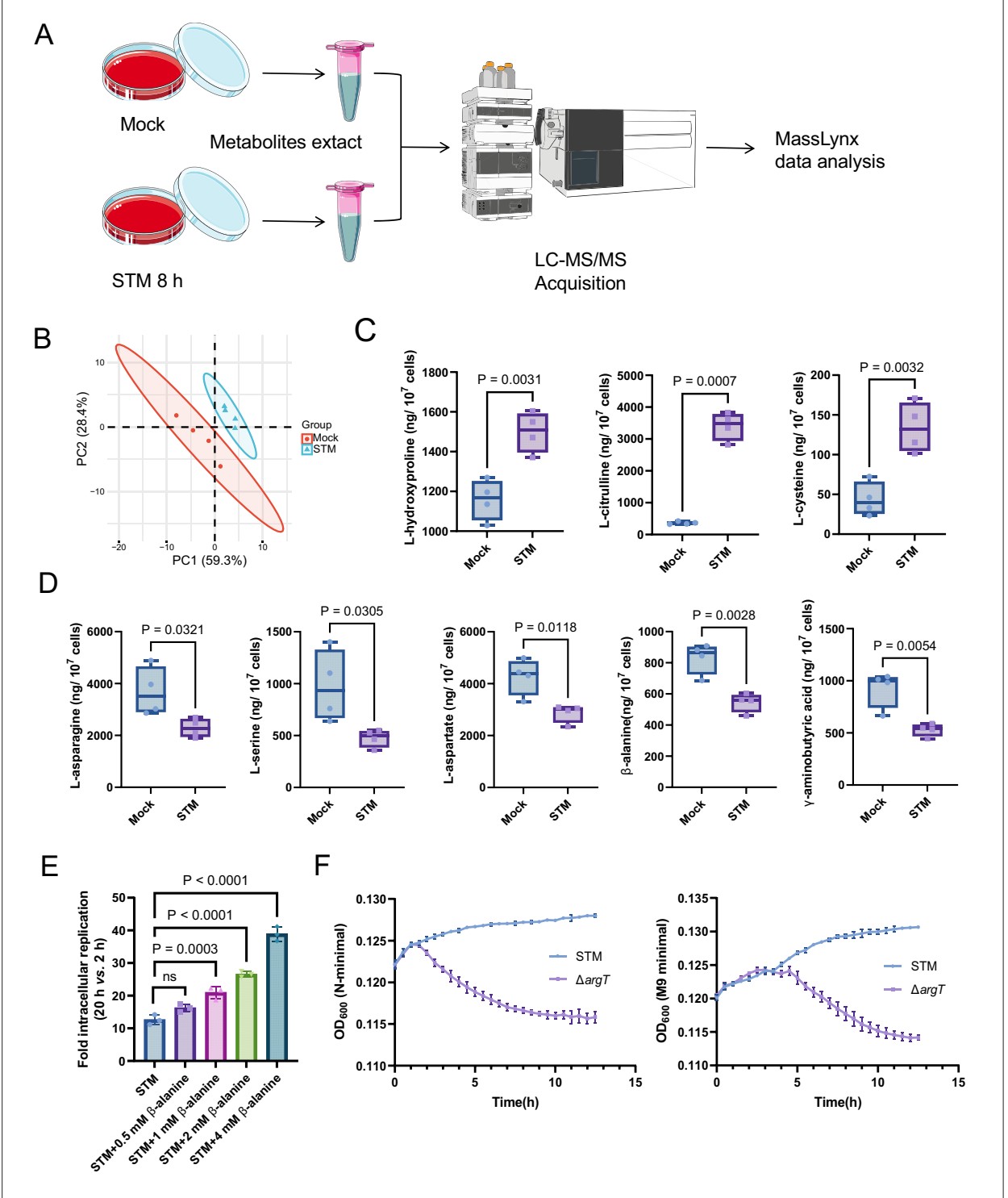

**Figure 1.** Host-derived β-alanine promotes *Salmonella* replication inside macrophages. (**A**) Schematic workflow for targeted metabolomics investigation of mock- and *Salmonella*-infected (STM) mouse RAW264.7 macrophages. Picture materials were used from bioicons (https://bioicons.com/). (**B**) Principal component analysis (PCA) score plots of metabolic profiles in the mock- and *Salmonella*-infected (STM) groups (n=4 biologically independent samples). (**C**) The concentrations of upregulated amino acids in the mock- and *Salmonella*-infected groups (n=4 biologically independent samples). (**D**) The concentrations of downregulated amino acids in the mock- and *Salmonella*-infected groups (n=4 biologically independent samples). (**E**) Fold intracellular replication (20 hr *vs.* 2 hr) of *Salmonella* WT in RAW264.7 cells in the presence of 0.5, 1, 2, 4 mM β-alanine. Data are presented as the mean ± SD, n=3 independent experiments. (**F**) Growth curves of *Salmonella* WT and the *argT* mutant (Δ*argT*) in N-minimal (left) and M9 minimal (right)

*Figure 1 continued on next page*

*Figure 1 continued*

medium supplemented with β-alanine (1 mM) as the sole carbon source. Data are presented as mean ± SD, n=4 independent experiments. Statistical significance was assessed using two-sided Student's *t*-test (**C, D**) and one-way ANOVA (**E**).

The online version of this article includes the following source data and figure supplement(s) for figure 1:

**Source data 1.** The numerical source data corresponds to *Figure 1*.

**Figure supplement 1.** The levels of reactive oxygen species (ROS) and reactive nitrogen species (RNS) in RAW264.7 cells after infection with *Salmonella* WT for 8 hr, in the presence or absence of 1 mM β-alanine.

**Figure supplement 1—source data 1.** The numerical source data corresponds to *Figure 1—figure supplement 1*.

**Figure supplement 2.** Flow cytometry analysis was conducted to determine the percentage of pro-inflammatory M1 macrophages (CD86+) and anti-inflammatory M2 macrophages (CD163+).

**Figure supplement 2—source data 1.** The numerical source data corresponds to *Figure 1—figure supplement 2*.

**Figure supplement 3.** Growth curves of *Salmonella* wild-type (WT) and the *cycA* mutant (ΔcycA) in N-minimal medium supplemented with β-alanine (1 mM) as the sole carbon source.

**Figure supplement 3—source data 1.** The numerical source data corresponds to *Figure 1—figure supplement 3*.

**Figure supplement 4.** Growth curves of *Salmonella* wild-type (WT) and the *cycA* mutant (ΔcycA) in M9 minimal medium supplemented with β-alanine (1 mM) as the sole carbon source.

**Figure supplement 4—source data 1.** The numerical source data corresponds to *Figure 1—figure supplement 4*.

**Figure supplement 5.** Fold intracellular replication (20 hr *vs.* 2 hr) of *Salmonella* wild-type and ΔcycA in RAW264.7 cells.

**Figure supplement 5—source data 1.** The numerical source data corresponds to *Figure 1—figure supplement 5*.

**Figure supplement 6.** Liver and spleen bacterial burdens, and body weight of mice infected with *Salmonella* wild-type (STM) and *cycA* mutant (ΔcycA), at day 3 post-infection.

**Figure supplement 6—source data 1.** The numerical source data corresponds to *Figure 1—figure supplement 5*.

**Figure supplement 7.** Growth curves of *Salmonella* wild-type (WT) and the *gabP* mutant (ΔgabP) in N-minimal medium supplemented with β-alanine (1 mM) as the sole carbon source.

**Figure supplement 7—source data 1.** The numerical source data corresponds to *Figure 1—figure supplement 5*.

**Figure supplement 8.** Growth curves of *Salmonella* wild-type (WT) and the *gabP* mutant (ΔgabP) in M9 minimal medium supplemented with β-alanine (1 mM) as the sole carbon source.

**Figure supplement 8—source data 1.** The numerical source data corresponds to *Figure 1—figure supplement 5*.

β-alanine (*Figure 1E*). Furthermore, β-alanine enhanced *Salmonella* intracellular replication in a dose-dependent manner (*Figure 1E*). The results suggest that host-derived β-alanine facilitates *Salmonella* replication inside macrophages. We then investigated whether β-alanine-mediated *Salmonella* growth promotion is due to the changes in antimicrobial activity of the macrophages. We observed that the addition of 1 mM β-alanine did not influence the ROS (reactive oxygen species) and RNS (reactive nitrogen species) levels in *Salmonella*-infected RAW264.7 cells (*Figure 1—figure supplement 1*). Flow cytometry analysis indicated that the addition of 1 mM β-alanine did not affect the percentage of pro-inflammatory M1 macrophages (CD86+) and anti-inflammatory M2 macrophages (CD163+) during *Salmonella* infection (*Figure 1—figure supplement 2*), implying that the addition of β-alanine to macrophages does not change their immune response. Combining these results, we can further infer that *Salmonella* use host-derived β-alanine for intracellular growth.

Direct validation of *Salmonella* using host-derived β-alanine for intracellular growth requires a mutant that has a defect in β-alanine uptake. *Escherichia coli* uptakes β-alanine via the transporter protein CycA (*Schneider et al., 2004*). However, the *Salmonella* ΔcycA mutant was able to use β-alanine as the sole carbon source for growth in minimal medium (*Figure 1—figure supplements 3 and 4*), indicating that CycA is not a transporter for β-alanine in *Salmonella*. Consistent with these results, mutation of *cycA* did not influence the replication of *Salmonella* in RAW264.7 cells (*Figure 1—figure supplement 5*) or colonization in mouse systemic tissues (liver and spleen; *Figure 1—figure supplement 6*). In *E. coli*, GabP transports γ-aminobutyric acid (GABA), a structural analog of β-alanine, and may also transport β-alanine (*Pavić et al., 2021*). Nevertheless, the *Salmonella* ΔgabP mutant displayed no growth defect in minimal medium with β-alanine as the sole carbon source (*Figure 1—figure supplements 7 and 8*), indicating that GabP is not involved in β-alanine uptake in *Salmonella*. Strikingly, the ΔargT mutant—defective in arginine uptake—showed markedly decreased growth in

the minimal medium with β-alanine as the sole carbon source (*Figure 1F*), suggesting that ArgT also transports β-alanine in *Salmonella*. It has been reported that ArgT is essential for *Salmonella* replication within macrophages and full virulence in vivo (*Das et al., 2010*). Given that ArgT is involved in both arginine and β-alanine uptake (as verified in this study), whether the attenuated virulence of the Δ*argT* mutant is due to a deficiency in β-alanine or arginine requires further investigation.

## De novo β-alanine synthesis is critical for *Salmonella* replication inside macrophages

*Salmonella* can de novo synthesize β-alanine *via* the decarboxylation of L-aspartate, which is catalyzed by L-aspartate decarboxylase (PanD) (*Figure 2A*) and is reportedly the rate-limiting step of β-alanine generation (*Begley et al., 2001*; *Schmitzberger et al., 2003*; *West et al., 1985*). To further assess the role of β-alanine in *Salmonella* intracellular replication, we analyzed the expression level of the *Salmonella panD* gene in macrophages and the impact of *panD* mutation on the ability of *Salmonella* to replicate in macrophages. Quantitative real-time PCR (qRT–PCR) assays revealed that the expression level of *panD* was significantly (p<0.01) greater in RAW264.7 cells than in RPMI-1640 medium (*Figure 2B*). Increased expression of *panD* was also observed in N-minimal medium, a widely used medium that mimics the conditions inside macrophages, as revealed by qRT–PCR and bioluminescent reporter assays (*Figure 2C and D*). These results demonstrate that *panD* expression is enhanced during *Salmonella* growth inside macrophages, suggesting a relationship between *panD* expression and intracellular *Salmonella* growth.

We then constructed the *panD* mutant strain Δ*panD* and compared the replication ability of the Δ*panD* strain and the *Salmonella* Typhimurium 14,028 s wild-type (WT) strain in RAW264.7 cells. Gentamicin protection assays showed that the replication of Δ*panD* in RAW264.7 cells decreased 2.4-fold at 20 hr post-infection compared with that of the WT strain (p<0.001), while complementation of Δ*panD* with the *panD* gene restored the replication ability of the mutant strain in RAW264.7 cells (*Figure 2E*). Immunofluorescence analysis revealed that the number of Δ*panD* in each infected RAW264.7 cell was comparable to that of the WT strain at the initial infection stage (2 hr), but at 20 hr post-infection, the number of Δ*panD* in each infected RAW264.7 cell was significantly (p<0.0001) lower than that of the WT strain (*Figure 2F and G*). These results indicate that *panD* contributes to *Salmonella* replication in macrophages. The growth rates of Δ*panD* in LB medium and RPMI medium resembled those of the WT (*Figure 2—figure supplements 1 and 2*), indicating that the impaired intracellular replication ability of the mutant was not due to a growth defect. Moreover, the replication defect of Δ*panD* in RAW264.7 cells was relieved by the addition of 1 mM β-alanine to the RPMI medium (*Figure 2H*). Furthermore, we examined the role of β-alanine synthesis in the intracellular replication of *Salmonella* within another typical serovar, *Salmonella enterica* serovar Typhi (*S.* Typhi), a serovar specific to humans and the causative agent of typhoid fever (*de Jong et al., 2012*). We found that the replication of *S.* Typhi Δ*panD* in human THP-1 macrophages was reduced by 2.6-fold compared to the *S.* Typhi Ty2 WT strain (p<0.01) (*Figure 2—figure supplement 3*), suggesting that *panD* also facilitates *S.* Typhi replication within human macrophages.

These data collectively suggest that β-alanine synthesis is critical for *Salmonella* replication inside macrophages.

## De novo β-alanine synthesis is critical for systemic *Salmonella* infection in mice

As replication in macrophages is a key determinant of systemic *Salmonella* infection, we reasoned that β-alanine synthesis also influences *Salmonella* systemic infection in vivo. To determine whether β-alanine influences systemic *Salmonella* infection, we conducted mouse infection assays using intraperitoneal (i.p.) injection. This method allows *Salmonella* to disseminate directly to systemic sites *via* the lymphatic and bloodstream systems, bypassing the need for intestinal invasion (*Pandeya et al., 2023*; *Silva et al., 2016*). BALB/c mice were infected by i.p. injection of 5000 CFU of WT, Δ*panD*, or the complemented strain c*panD*. The survival rate, body weight, bacterial burden in the liver and spleen, and liver histopathological alterations of the infected mice were measured (*Figure 3A*). The WT-infected mice exhibited high lethality and marked loss of body weight within 5 d, and all mice died within 9 d of infection (*Figure 3B and C*). In contrast, the Δ*panD*-infected mice displayed significantly improved survival rates and body weights, and no mice died within the 10 d surveillance period

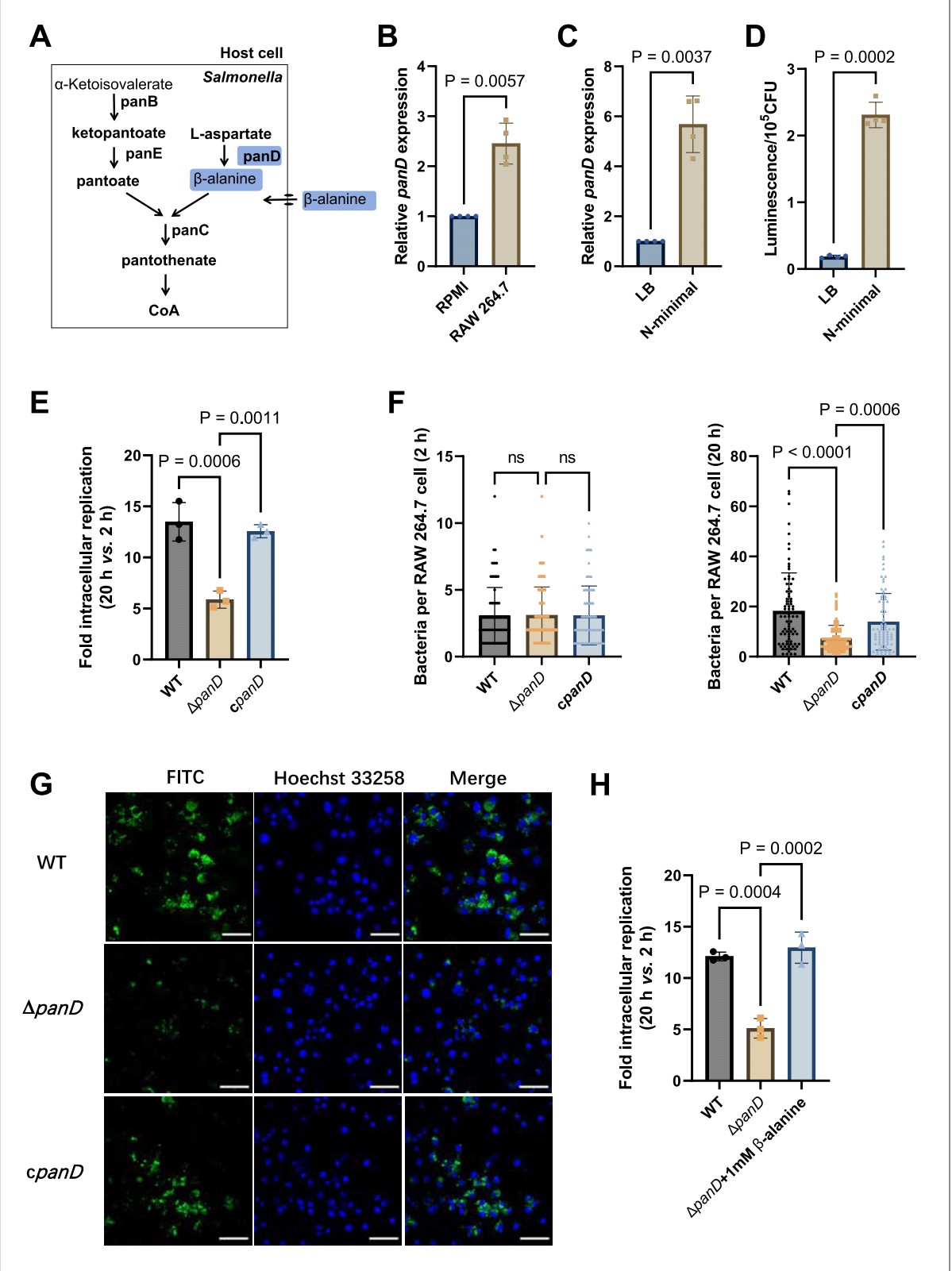

**Figure 2.** De novo β-alanine synthesis is critical for *Salmonella* replication inside macrophages. (**A**) Scheme of β-alanine and the downstream CoA biosynthesis pathway in *Salmonella*. (**B**) Quantitative real-time PCR (qRT–PCR) analysis of the expression of the *Salmonella panD* gene in RAW264.7 cells (8 hr post-infection) and RPMI-1640 medium. (**C**) qRT–PCR analysis of the expression of the *Salmonella panD* gene in N-minimal medium and LB medium. (**D**) Expression of the *panD*-lux transcriptional fusion in N-minimal medium and LB medium. Luminescence values were normalized to $10^5$

*Figure 2 continued on next page*

*Figure 2 continued*

bacterial CFUs. (**E**) Fold intracellular replication (20 hr *vs.* 2 hr) of *Salmonella* Typhimurium 14,028 s wild-type (WT), the *panD* mutant (Δ*panD*), and the complemented strain (c*panD*) in RAW264.7 cells. (**F**) Number of intracellular *Salmonella* WT, Δ*panD*, and c*panD* strains per RAW264.7 cell at 2 and 20 hr post-infection. The number of intracellular bacteria per infected cell was estimated in random fields, n=80 cells per group from three independent experiments. (**G**) Representative immunofluorescence images of *Salmonella* WT, Δ*panD*, and c*panD* in RAW264.7 cells at 20 hr post-infection (green, *Salmonella*; blue, nuclei; scale bars, 50 μm). Images are representative of three independent experiments. (**H**) Replication of *Salmonella* WT and Δ*panD* in RAW264.7 cells in the presence or absence of 1 mM β-alanine. The data are presented as the mean ± SD, n=3 (**B–E**, **H**) independent experiments. Statistical significance was assessed using a two-sided Student's *t*-test (**B–D**) or one-way ANOVA (**E, F, H**). ns, not Significant.

The online version of this article includes the following source data and figure supplement(s) for figure 2:

**Source data 1.** The numerical source data corresponds to *Figure 2*.

**Figure supplement 1.** Growth curves of *Salmonella* wild-type (WT), *panD* mutant (Δ*panD*), and the complemented strain (c*panD*) in LB medium.

**Figure supplement 1—source data 1.** The numerical source data corresponds to *Figure 2—figure supplement 1*.

**Figure supplement 2.** Growth curves of *Salmonella* wild-type (WT), *panD* mutant (Δ*panD*) and the complemented strain (c*panD*) in RPMI-1640 medium (**B**).

**Figure supplement 2—source data 1.** The numerical source data corresponds to *Figure 2—figure supplement 2*.

**Figure supplement 3.** Fold intracellular replication (20 hr *vs.* 2 hr) of *Salmonella enterica* serovar Typhi Ty2 wild-type (WT), the *panD* mutant (Δ*panD*) in human THP-1 monocyte-like cell line (ATCC TIB-22).

**Figure supplement 3—source data 1.** The numerical source data corresponds to *Figure 2—figure supplement 3*.

(*Figure 3B and C*). Consistent with these results, the bacterial burden in the liver and spleen of the Δ*panD*-infected mice was significantly decreased, and the body weight was significantly increased compared with that of the WT-infected mice on day 3 post-infection (*Figure 3D*). Complementation of Δ*panD* with *panD* significantly decreased the survival rate and body weight of infected mice but significantly increased the bacterial burden in the liver and spleen of the infected mice (*Figure 3B–D*). Through immunofluorescence staining, we examined the bacterial count in liver macrophages of mice infected with WT, Δ*panD*, and complemented strain. The results showed that the bacterial count in each macrophage from Δ*panD*-infected mice was significantly (*P*<0.0001) lower than that in WT-infected mice, on day 5 post-infection. Complementation of Δ*panD* with *panD* restored the bacterial count in each macrophage to WT level (*Figure 3E*). Furthermore, H&E staining revealed increased aggregation of inflammatory cells and pyknosis in the livers of the WT-infected mice on day 5 post-infection, while these histopathological alterations were obviously reduced in the livers of Δ*panD*-infected mice (*Figure 3F*). Taken together, these results reveal that β-alanine synthesis is critical for systemic *Salmonella* infection in mice.

## β-Alanine is involved in the regulation of several metabolic pathways in *Salmonella*

To explore the mechanism(s) associated with β-alanine-mediated promotion of *Salmonella* replication in macrophages and in vivo virulence, we performed RNA sequencing (RNA-seq) to reveal the differences in gene transcripts between *Salmonella* WT and Δ*panD* strains grown in N-minimal medium. PCA plot of the global transcriptomic profiles clearly demonstrated separation between the WT and Δ*panD* strains (*Figure 4A*). Remarkable transcriptional changes were observed due to the mutation of *panD*. Compared with those in the WT strain, 1379 genes were differentially expressed in the Δ*panD* strain, with 561 upregulated genes and 618 downregulated genes (fold change ≥2 and *p*-value <0.05; *Figure 4B*). Gene Ontology (GO) enrichment analysis revealed that the differentially expressed genes (DEGs) were mainly involved in the metabolism and biosynthesis of several amino acids (including arginine, leucine, histidine, and branched amino acids), carboxylic acid metabolism, small molecule biosynthesis, and aerobic respiration (*Figure 4C*). Kyoto Encyclopedia of Genes and Genomes (KEGG) pathway enrichment analysis also revealed a high frequency of terms related to metabolism, including amino acid metabolism, lipid metabolism, carbohydrate metabolism, energy metabolism, and nucleotide metabolism (*Figure 4D*). These data collectively indicate that β-alanine is involved in the regulation of a series of metabolic pathways in *Salmonella*.

Further analysis of the downregulated DEGs (activated by PanD) revealed that mutation of *panD* decreased the expression of genes involved in even pathways that are associated with the virulence of *Salmonella* or other bacterial pathogens, including methionine metabolism, fatty acid β-oxidation,

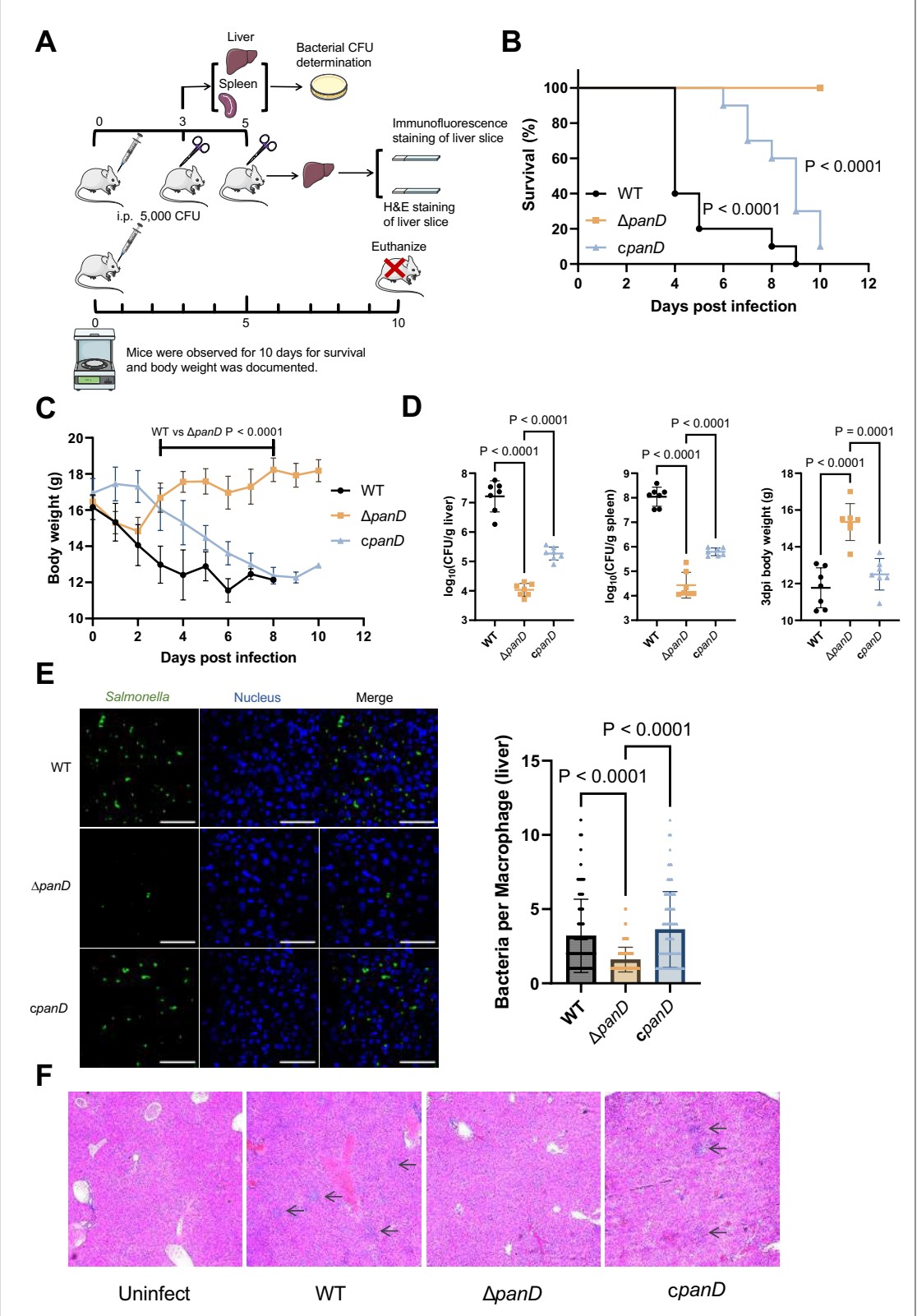

**Figure 3.** De novo β-alanine synthesis is critical for systemic *Salmonella* infection in mice. (**A**) Schematic illustration of the mouse infection assays. Picture materials were used from bioicons (https://bioicons.com/). (**B, C**) Survival curves (**B**) and body weight dynamics (**C**) of mice infected i.p. with *Salmonella* wild-type (WT), Δ*panD*, or c*panD*. n=10 randomly assigned mice per group. (**D**) Liver and spleen bacterial burdens and body weights of mice infected with *Salmonella* WT, Δ*panD*, or c*panD* on day 3 post-infection. n=7 randomly assigned mice per group. (**E**) Representative immunofluorescence

*Figure 3 continued on next page*

*Figure 3 continued*

images and intracellular bacterial counts of *Salmonella* WT, Δ*panD*, and c*panD* in mouse liver at 5 d post-infection (green, *Salmonella*; blue, nuclei; scale bars, 50 μm). Images are representative of three independent experiments. The number of intracellular bacteria per infected cell was estimated in random fields, with n=80 cells per group from three independent experiments. (**F**) Representative H&E-stained liver sections from mice that were left uninfected or infected with *Salmonella* WT, Δ*panD*, or c*panD* on day 5 post-infection. Arrows indicate severe inflammatory cell infiltration in the mouse liver. Images are representative of three independent experiments. The data are presented as the mean ± SD (**B–E**). Statistical significance was assessed using the log-rank Mantel–Cox test (**B**), two-sided Student's *t*-test (**C**), or one-way ANOVA (**D, E**).

The online version of this article includes the following source data for figure 3:

**Source data 1.** The numerical source data corresponds to *Figure 3*.

histidine biosynthesis, and the transport of zinc, galactose, potassium, and polyamine (*Figure 4E*). Zinc and potassium uptake are associated with the virulence of *Salmonella* (zinc acquisition promotes *Salmonella* Typhimurium virulence in mice, and potassium acquisition promotes *Salmonella* Enteritidis virulence in chickens) (*Ammendola et al., 2007*; *Battistoni et al., 2017*; *Ilari et al., 2016*; *Liu et al., 2013*), while the remaining 5 pathways are involved in the pathogenicity of other pathogens (*Basavanna et al., 2013*; *Paiva et al., 2016*; *Feldman et al., 2016*; *Lauriano et al., 2004*; *Martínez-Guitián et al., 2019*). In addition, the expression of the LysR-type transcriptional regulator LeuO, which activates the expression of the *leuABCD* leucine synthesis operon and numerous virulence genes in *Salmonella* Typhimurium (*Dillon et al., 2012*; *Guadarrama et al., 2014*; *Hernández-Lucas et al., 2008*), was also downregulated in the Δ*panD* strain (*Figure 4E*). In line with the decreased expression of *leuO*, the expression of *leuABCD* was downregulated in the Δ*panD* strain (*Figure 4E*).

We selected 16 downregulated DEGs (including the regulatory gene *leuO* and genes from the above 7 pathways) for qRT–PCR analysis. The results showed that the expression of all 16 genes significantly (p<0.05) decreased in the Δ*panD* mutant compared with the WT strain (*Figure 4F*), and complementation of Δ*panD* with *panD* restored the gene expression to the WT level (*Figure 4F*), thus confirming the positive regulation of these pathways and LeuO by β-alanine.

*Salmonella* pathogenesis largely depends on virulence genes encoded by *Salmonella* pathogenicity islands (SPIs), with SPI-1 to SPI-5 being well-characterized for their involvement in *Salmonella* virulence (*Han et al., 2024*). SPI-2 gene expression is essential for *Salmonella* replication in macrophages and systemic infection (*Deng et al., 2017*), yet the expression of SPI-2 genes was unaffected by *panD* mutation (*Figure 4—figure supplement 1*). Moreover, the gene expression of four other virulence-associated SPIs, SPI-1, SPI-3, SPI-4, and SPI-5, is also unaffected by *panD* mutation (*Figure 4—figure supplements 2 and 3*).

Taken together, these data suggest that β-alanine might promote *Salmonella* intracellular replication and virulence by activating virulence-associated pathway(s) or activating the virulence-associated regulator LeuO, rather than by activating the expression of virulence genes encoded within pathogenicity islands.

## β-alanine promotes *Salmonella* virulence in vivo partially by increasing the expression of zinc transporter genes

Next, we inactivated the seven downregulated pathways mentioned above, as well as the regulatory gene *leuO* in *Salmonella,* to uncover the mechanism(s) by which β-alanine promotes *Salmonella* virulence in vivo. Mouse infection assays revealed that mutations in *fadAB*, *metR*, *hisABCDFGHL*, *kdpABC*, *mglABC*, and *potFGHI*, which are associated with fatty acid β-oxidation, methionine metabolism, histidine biosynthesis, potassium uptake, galactose uptake, and polyamine uptake, respectively, did not influence *Salmonella* colonization in the mouse liver or spleen (*Figure 5A and B*) or the body weight of infected mice (*Figure 5C*). Interestingly, although LeuO has been reported to be associated with the regulation of a diverse set of virulence factors (*Dillon et al., 2012*; *Guadarrama et al., 2014*), mutation of the regulatory gene *leuO* did not influence *Salmonella* colonization in the mouse liver or spleen (*Figure 5A and B*) or the body weight of infected mice (*Figure 5C*). In contrast, mutation of the zinc transporter gene *znuA* significantly decreased *Salmonella* colonization in the mouse liver and spleen (*Figure 5D*, left and middle panels); this result is consistent with previous studies (*Ammendola et al., 2007*; *Battistoni et al., 2017*; *Ilari et al., 2016*). Accordingly, the body weight of Δ*znuA*-infected mice was significantly (p<0.001) greater than that of the WT-infected mice (*Figure 5D*, right panel). These results indicate that β-alanine might promote *Salmonella* virulence in vivo by promoting zinc uptake.

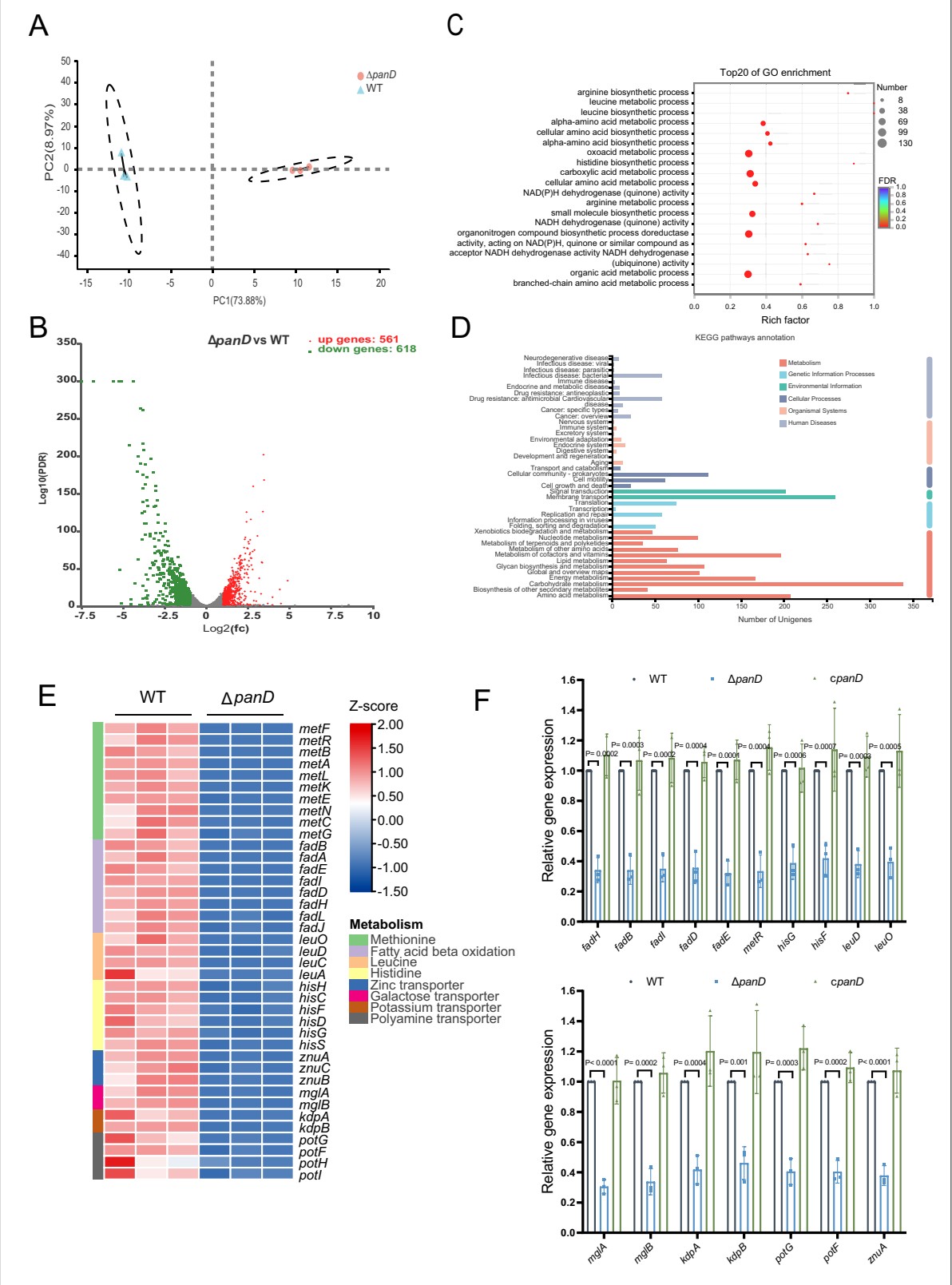

**Figure 4.** β-Alanine is involved in the regulation of several metabolic pathways in *Salmonella*. (**A**) Principal component analysis (PCA) score plots of transcriptomic profiles of *Salmonella* wild-type (WT) and Δ*panD* (n=3 biologically independent samples). (**B**) Volcano plot of the differentially expressed genes (DEGs) in *Salmonella* WT versus Δ*panD*. The upper right section (red dots) indicates the upregulated DEGs, and the upper left section (green dots) indicates the downregulated DEGs. (**C**) Gene Ontology (GO) enrichment analysis of DEGs. Bubble chart showing the top 20

*Figure 4 continued on next page*

*Figure 4 continued*

enriched Gene Ontology (GO) terms. (**D**) Kyoto Encyclopedia of Genes and Genomes (KEGG) pathway enrichment analysis of DEGs. (**E**) Expression of the downregulated pathways (activated by PanD) is shown in the Z score-transformed heatmap, with red representing higher abundance and blue representing lower abundance. (**F**) Quantitative real-time PCR (qRT–PCR) analysis of the mRNA levels of 16 selected downregulated DEGs in *Salmonella* WT, Δ*panD*, and c*panD*. The data are presented as the mean ± SD, n=3 independent experiments. Statistical significance was assessed using two-way ANOVA.

The online version of this article includes the following source data and figure supplement(s) for figure 4:

**Source data 1.** The numerical source data corresponds to *Figure 4*.

**Figure supplement 1.** Expression of SPI-2 is shown in the Z-score transformed heatmap, with orange representing higher and blue representing lower abundance.

**Figure supplement 1—source data 1.** The numerical source data corresponds to *Figure 4—figure supplement 1*.

**Figure supplement 2.** Expression of SPI-1 is shown in the Z-score transformed heatmap, with orange representing higher and blue representing lower abundance.

**Figure supplement 2—source data 1.** The numerical source data corresponds to *Figure 4—figure supplement 2*.

**Figure supplement 3.** Expression of SPI-3, SPI-4, and SPI-5 genes is shown in the Z-score transformed heatmap, with orange representing higher and blue representing lower abundance.

**Figure supplement 3—source data 1.** The numerical source data corresponds to *Figure 4—figure supplement 3*.

To test this hypothesis, we initially evaluated the zinc content in the livers of WT- and Δ*panD*-infected mice, on day 3 post-infection. We observed that the zinc concentration in the macrophages of Δ*panD*-infected mouse livers was 3.2-fold higher than in those of WT-infected mice (p<0.0001; *Figure 5E*), suggesting that the *panD* gene and β-alanine are crucial for *Salmonella* to obtain zinc from host cells.

Next, we constructed a double mutant, Δ*panD*Δ*znuA*, and compared colonization of the mouse liver and spleen of the double mutant to that of the single mutant, Δ*znuA*. The results showed that colonization of the liver and spleen of infected mice by Δ*panD*Δ*znuA* was significantly lower than that of infected mice colonized by Δ*znuA* (*Figure 5D*, left and middle panels). In agreement with these results, the body weight of Δ*panD*Δ*znuA*-infected mice was greater than that of Δ*znuA*-infected mice (*Figure 5D*, right panel), suggesting that the contribution of *panD* to the virulence of *Salmonella* is partially dependent on *znuA*.

Collectively, these data indicate that β-alanine promotes in vivo virulence of *Salmonella* partially by increasing the expression of zinc transporter genes.

## β-alanine promotes *Salmonella* replication within macrophages partially by increasing the expression of zinc transporter genes

To determine whether β-alanine influences *Salmonella* intracellular replication by acting on zinc transporters, the zinc content in RAW 264.7 macrophages infected with WT and Δ*panD* was also examined. We observed that the zinc concentration in Δ*panD*-infected RAW 264.7 cells increased by 1.8-fold compared to WT-infected cells (p<0.0001; *Figure 6A*), further indicating the *panD* gene and β-alanine are crucial for *Salmonella* to absorb zinc from macrophages.

We then analyzed the ability of Δ*znuA* and Δ*panD*Δ*znuA* to replicate in RAW264.7 macrophages *via* gentamicin protection assays. The results showed that the replication of Δ*panD*Δ*znuA* in RAW264.7 cells was significantly reduced compared with that of the single mutant Δ*znuA* (*Figure 6B*), implying that the contribution of *panD* to the intracellular replication of *Salmonella* is partially dependent on *znuA*. The addition of 100 µM zinc to RPMI medium increased the replication of Δ*panD* in RAW264.7 cells (*Figure 6C*), while the addition of 1 mM β-alanine to RPMI medium did not increase the replication of Δ*znuA* (*Figure 6D*), suggesting that the impaired replication due to the decrease in β-alanine can be relieved by zinc supplementation.

Taken together, these data indicate that β-alanine promotes *Salmonella* replication within macrophages by increasing the expression of zinc transporter genes.

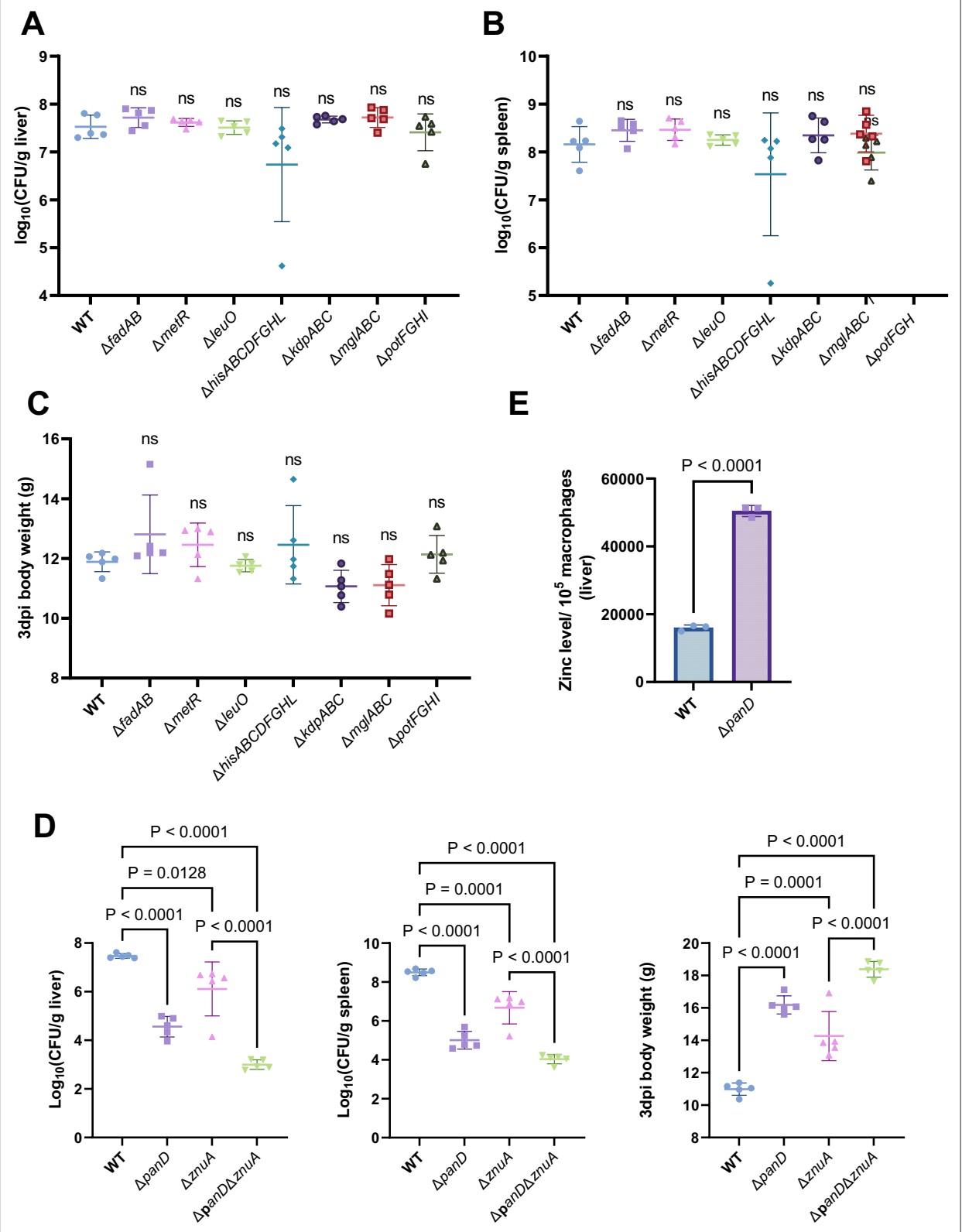

**Figure 5.** β-alanine promotes *Salmonella* virulence in vivo partially by increasing the expression of zinc transporter genes. (**A, B, C**) Liver (**A**) and spleen (**B**) bacterial burdens and body weight (**C**) of mice infected with *Salmonella* wild-type (WT), Δ*fadAB*, Δ*metR*, Δ*hisABCDFGHL*, Δ*kdpABC*, Δ*mglABC*, Δ*potFGHI*, or Δ*leuO* on day 3 post-infection. n=5 mice per group. (**D**) Liver and spleen bacterial burdens and body weights of mice infected with *Salmonella* WT, Δ*panD*, Δ*znuA* or Δ*panD*Δ*znuA* on day 3 post-infection. n=5 mice per group. (**E**) The zinc levels in the livers of mice infected with either

*Figure 5 continued on next page*

*Figure 5 continued*

*Salmonella* WT or Δ*panD* for 3 d, n=5 mice per group. The data are presented as the mean ± SD (A–E). Statistical significance was assessed using one-way ANOVA (**A-D**), two-sided Student's *t*-test (**E**). ns, not Significant.

The online version of this article includes the following source data for figure 5:

**Source data 1.** The numerical source data corresponds to *Figure 5*.

## Discussion

Replication within host macrophages is a crucial step for *Salmonella* to cause life-threatening systemic infection in the host (*Bomjan et al., 2019*; *Lathrop et al., 2015*), while the crosstalk between *Salmonella* and macrophages at the metabolic interface is critical for intracellular *Salmonella* replication (*Dandekar et al., 2014*; *Herrero-Fresno and Olsen, 2018*; *Lynch and Lesser, 2021*; *Rosenberg et al., 2021*; *Thompson et al., 2018*). Emerging evidence suggests that several metabolites affect the replication of *Salmonella* in macrophages. The promotion of intracellular replication by metabolites is possibly achieved in three ways: (i) metabolites are utilized by *Salmonella* as nutrients for intracellular growth (*Bowden et al., 2009*; *Eisenreich et al., 2010*; *Wang et al., 2021a*); (ii) *Salmonella* senses metabolites as environmental cues to activate the expression of virulence genes (*Jiang et al., 2021*; *Wang et al., 2023*); and (iii) metabolites can regulate the immune responses of macrophages (*Michelucci et al., 2013*; *Peace and O'Neill, 2022*; *Yang and Cong, 2021*). In this study, we demonstrated that *Salmonella* promotes its replication inside macrophages by utilizing both host- and bacterial-derived β-alanine (*Figure 6E*). We showed that β-alanine promotes *Salmonella* intracellular replication and systemic infection partially by increasing the expression of zinc transporter genes and therefore, the uptake of zinc by intracellular *Salmonella* (*Figure 6E*). Therefore, this work identified another metabolite that can influence the replication of *Salmonella* in macrophages and illustrated the mechanism by which β-alanine promotes intracellular *Salmonella* replication.

We observed that *Salmonella*-infected macrophages contained lower β-alanine levels than mock-infected macrophages, while β-alanine supplementation in the cell medium increased the replication of *Salmonella* in macrophages, revealing that *Salmonella* uptakes host-derived β-alanine to promote intracellular replication. In addition, a deficiency in the biosynthesis of β-alanine (*via* mutation of the rate-limiting gene *panD*) reduced *Salmonella* replication in macrophages and systemic infection in mice, suggesting that *Salmonella* also utilizes bacterial-derived β-alanine to promote intracellular replication and pathogenicity. It is known that bacteria are quite stringent with their energy resources (*Bergkessel, 2020*; *Bosdriesz et al., 2015*), while the results of this work indicate that either acquisition from the host or de novo synthesis of β-alanine is critical for *Salmonella* replication inside macrophages. We speculate that *Salmonella* relies on a large amount of β-alanine to efficiently replicate in macrophages, thereby highlighting the importance of β-alanine for *Salmonella* intracellular growth. Nevertheless, unlike the closely related species *E. coli*, which takes up β-alanine via the transporter protein CycA (*Schneider et al., 2004*), *Salmonella* does not use CycA to uptake β-alanine. GabP, the GABA transporter, is potentially involved in the uptake of β-alanine in *E. coli* (*Pavić et al., 2021*). However, *Salmonella* does not utilize GabP to uptake β-alanine either. Ultimately, we revealed that ArgT, the transporter of arginine, is involved in the transport of β-alanine in *Salmonella*. ArgT has been reported to be essential for *Salmonella* replication within macrophages and for full virulence in vivo (*Das et al., 2010*). However, the attenuated virulence of the Δ*argT* mutant due to a deficiency in β-alanine or arginine requires further investigation.

Several amino acids, including lysine, proline, arginine, aspartate, and asparagine, have previously been reported to be associated with the pathogenicity of *Salmonella* (*Popp et al., 2015*; *Steeb et al., 2013*). These amino acids are involved in the synthesis of proteins in *Salmonella*. In contrast, β-alanine is not incorporated into proteins but can participate in the regulation of bacterial activity through the synthesis of pantothenate and CoA (*Webb et al., 2004*; *White et al., 2001*; *Yuan et al., 2022*). Accordingly, our transcriptome data showed that a deficiency in β-alanine biosynthesis affected the expression of *Salmonella* genes involved in a series of important metabolic pathways. Importantly, although β-alanine does not influence the gene expression of SPIs, it activates methionine metabolism; fatty acid β-oxidation; histidine biosynthesis; and the transport of zinc, galactose, potassium, and polyamine, which have been previously known to be associated with the virulence of *Salmonella* and other bacterial pathogens. Further analysis revealed that β-alanine promotes *Salmonella* intracellular

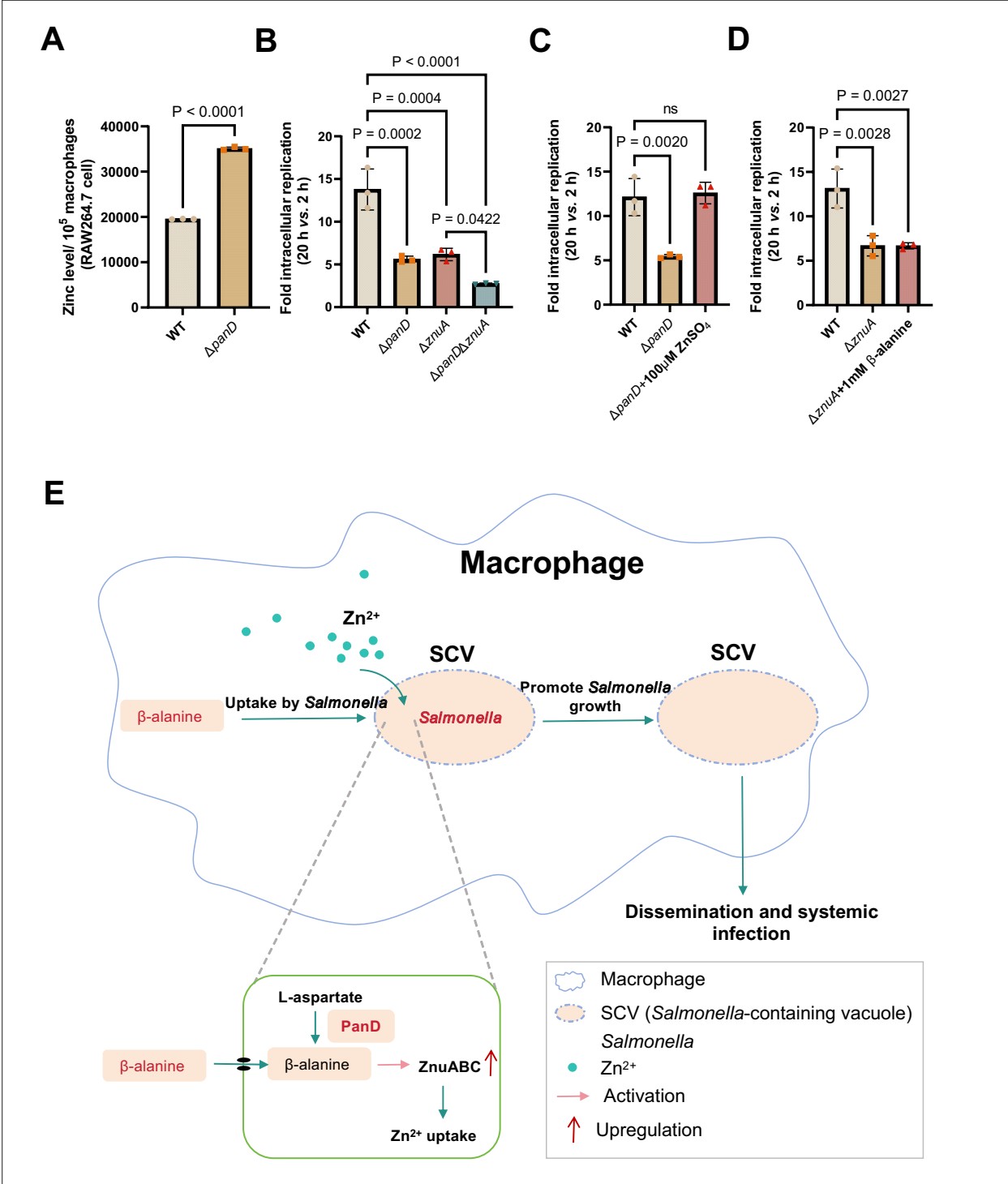

**Figure 6.** β-alanine promotes *Salmonella* replication within macrophages partially by increasing the expression of zinc transporter genes. (**A**) The zinc levels in RAW264.7 cells after infection with *Salmonella* wild-type (WT) or Δ*panD* for 8 hr. (**B**) Replication of *Salmonella* WT, Δ*panD*, Δ*znuA*, and Δ*panD*Δ*znuA* in RAW264.7 cells. (**C**) Replication of *Salmonella* WT and Δ*panD* in RAW264.7 cells in the presence or absence of 100 μM ZnSO₄. (**D**) Replication of *Salmonella* WT and Δ*znuA* in RAW264.7 cells in the presence or absence of 1 mM β-alanine. The data are presented as the mean ± SD, n=3 independent experiments (**A–D**). Statistical significance was assessed using a two-sided Student's *t*-test (**A**), one-way ANOVA (**B-D**). ns, not Significant. (**E**) Schematic model of β-alanine-mediated *Salmonella* replication inside macrophages. In macrophages, *Salmonella* acquires β-alanine both via the uptake of β-alanine from host macrophages and the de novo synthesis of β-alanine. β-alanine promotes the expression of zinc transporter genes *ZnuABC*, which facilitate the uptake of zinc by intracellular *Salmonella*, therefore, promote *Salmonella* replication in macrophages and subsequent systemic infection.

*Figure 6 continued on next page*

*Figure 6 continued*

The online version of this article includes the following source data for figure 6:

**Source data 1.** The numerical source data corresponds to *Figure 6*.

replication and systemic infection partially by promoting the uptake of zinc. As for the other virulence-related metabolic pathways activated by β-alanine, methionine metabolism, fatty acid β-oxidation, and histidine biosynthesis contribute to the virulence of *Streptococcus pneumoniae* (*Basavanna et al., 2013*), *Yersinia pestis* (*Feldman et al., 2016*), and *Acinetobacter baumannii* (*Martínez-Guitián et al., 2019*), respectively; the uptake of galactose and polyamine influences the pathogenicity of *Francisella tularensis* (*Lauriano et al., 2004*) and the avian pathogenic *Escherichia coli* (*Paiva et al., 2016*), respectively; and the uptake of potassium is associated with the virulence of *Salmonella* Enteritidis in chickens (*Liu et al., 2013*). However, blocking these pathways did not influence the systemic *Salmonella* Typhimurium infection in mice, implying that different bacterial pathogens adopt different virulence strategies to establish infection. Determining other mechanism(s) by which β-alanine promotes the intracellular replication and systemic infection of *Salmonella* requires further investigation.

We observed that β-alanine also activates the expression of the LysR-type transcriptional regulator LeuO, which is known to regulate the expression of a wide variety of *Salmonella* genes that impact the stress response and virulence (*Dillon et al., 2012*; *Guadarrama et al., 2014*; *Hernández-Lucas et al., 2008*). Typically, LeuO activates the synthesis of the quiescent porins OmpS1 and OmpS2, which are required for *Salmonella* virulence in mice (*De la Cruz et al., 2007*; *Fernández-Mora et al., 2004*; *Rodríguez-Morales et al., 2006*). Consistent with the positive regulation of OmpS1 and OmpS2 by LeuO, lack of *leuO* in *Salmonella* also attenuated virulence in a mouse model (*Rodríguez-Morales et al., 2006*). However, the attenuated phenotypes of the *leuO* mutant in mice were not evident after i.p. injection relative to oral infection, as a previous report showed that the competitive index for the *leuO* mutant indicated approximately 1000-fold reduced colonization in mouse systemic tissues after oral infection but much less reduced colonization after i.p. injection (less than 10-fold) (*Rodríguez-Morales et al., 2006*). These results imply that LeuO might be predominantly associated with the invasion and intestinal infection of *Salmonella* but is weakly implicated in *Salmonella* intracellular replication and systemic infection. Therefore, it is not surprising that lack of *leuO* did not significantly affect *Salmonella* colonization in the systemic tissues of mice after i.p. injection, as revealed by our results.

Zinc is an essential micronutrient for all living organisms and is used as a cofactor for various enzymes and proteins (*Bock et al., 2016*; *Ilari et al., 2016*; *Tan et al., 2004*). In bacteria, zinc-binding proteins account for approximately 5% of the bacterial proteome and play crucial roles in bacterial metabolism and virulence (*Andreini et al., 2006*). Knockout of the zinc transporter ZnuABC reduces the virulence of *Salmonella*, *Campylobacter jejuni*, *Haemophilus ducreyi*, *Moraxella*, and urinary tract pathogenic *Escherichia coli* (UPEC) in the host (*Ilari et al., 2016*). Moreover, zinc is also utilized by *Salmonella* to subvert the antimicrobial host defense of macrophages by inhibiting NF-κB activation and impairing NF-κB-dependent bacterial clearance (*Jennings et al., 2018*; *Wu et al., 2017*). Therefore, the efficient acquisition of zinc may be crucial for *Salmonella*'s survival and replication within macrophages, where zinc availability is limited (*Ammendola et al., 2007*; *Ilari et al., 2016*). It has been reported that *Salmonella* utilizes the high-affinity ZnuABC zinc transporter to optimize zinc availability in host cells (*Ammendola et al., 2007*). In this study, we found that β-alanine can increase the expression of the zinc transporter genes *znuABC*, which could represent an additional mechanism for the efficient zinc uptake of *Salmonella* in macrophages. The results demonstrate a correlation between *Salmonella* β-alanine utilization and zinc uptake during intracellular infection and provide evidence that β-alanine can influence the macrophage immune response by acting on zinc uptake.

Overall, our findings suggest a model in which *Salmonella* exploits host- and bacterial-derived β-alanine to efficiently replicate in host macrophages and cause systemic disease. We propose that *Salmonella* requires a large amount of β-alanine during intracellular infection. The utilization of β-alanine promotes *Salmonella* uptake of the essential micronutrient zinc, which was previously shown to be required for the metabolic needs of intracellular *Salmonella* and to subvert the antimicrobial defense of macrophages by *Salmonella*. These observations provide new insight into *Salmonella* pathogenesis and the crosstalk between *Salmonella* and macrophages during intracellular infection. Considering that the *panD* gene is present in the genomes of all *Salmonella* strains and that mutation

of *panD* markedly reduced *Salmonella* replication ability in macrophages, as well as virulence in the mouse model, this gene may be used as a potential target to control systemic *Salmonella* infection.

## Materials and methods

### Key resources table

| Reagent type (species) or resource | Designation | Source or reference | Identifiers | Additional information |
|---|---|---|---|---|
| Biological sample (*M. musculus*) | BALB/c mice | Beijing Vital River Laboratory Animal Technology | Cat# 213 | 6-wk-old female |
| Cell line (*M. musculus*) | RAW264.7 | ATCC | Cat# TIB-71; RRID:CVCL_0493 | mouse macrophage-like cell line |
| Antibody | FITC-conjugated anti-*Salmonella* antibody (Rabbit polyclonal) | Abcam | Cat# ab20320; RRID:AB_445509 | IF (1:100) |
| Antibody | anti-CD86 antibody (Rat monoclonal) | Abcam | Cat# ab119857; RRID:AB_10902800 | Flow Cyt (1:100) |
| Antibody | anti-CD163 antibody (Rabbit monoclonal) | Abcam | Cat# ab182422; RRID:AB_2753196 | Flow Cyt (1:100) |
| Antibody | goat anti-rat IgG H&L (Alexa Fluor 488) antibody | Abcam | Cat# ab150165; RRID:AB_2650997 | Flow Cyt (1:100) |
| Antibody | donkey anti-rabbit IgG H&L (Alexa Fluor 647) antibody | Abcam | Cat# ab150075; RRID:AB_2752244 | Flow Cyt (1:100) |
| Chemical compound, drug | β-alanine | Solarbio | Cat# A9770 | N/A |
| Commercial assay or kit | EASYspinPlus bacterial RNA rapid extraction kit | Aidlab | Cat# RN0802 | N/A |
| Commercial assay or kit | 2×RealStar Power SYBR qPCR Mix | Genstar | Cat# A304 | N/A |
| Commercial assay or kit | StarScript III RT Kit | Genstar | Cat# A232 | N/A |
| Commercial assay or kit | ROS fluorescence probe (BBoxiProbe O06) | Bestbio | Cat# BB-46051 | N/A |
| Commercial assay or kit | RNS fluorescence probe (BBoxiProbe O52) | Bestbio | Cat# BB-470567 | N/A |
| Commercial assay or kit | gentleMACS/Mouse Liver Dissociation Kit | Miltenyi Biotec | Cat# 130-105-807 | N/A |
| Commercial assay or kit | Zinc fluorescence probe (Zinquin ethyl ester) | MKBio | Cat# MX4516 | N/A |
| Commercial assay or kit | Hematoxylin-Eosin(HE) staining kit | Sangon Biotech | Cat# E607318 | N/A |
| Software, algorithm | GraphPad Prism 9.5.1 | GraphPad | RRID:SCR_002798 | http://www.graphpad.com/ |
| Software, algorithm | Bowtie 2 | Bowtie | RRID:SCR_016368 | https://bowtie-bio.sourceforge.net/bowtie2/index.shtml |
| Software, algorithm | Xcalibur 4.0 | Thermo Fisher | RRID:SCR_014593 | https://www.thermofisher.cn/order/catalog/product/OPTON-30965?SID=srch-srp-OPTON-30965 |
| Software, algorithm | ZEN 2.3 ((blue edition)) | Carl Zeiss | RRID:SCR_013672 | https://www.zeiss.com/microscopy/en/products/software/zeiss-zen.html |
| Software, algorithm | Image J | National Institutes of Health | RRID:SCR_003070 | https://imagej.nih.gov/ij/ |
| Other | DAPI stain | Invitrogen | Cat# 21490 | N/A |

### Ethics statement

Six-wk-old female BALB/c mice were obtained from Beijing Vital River Laboratory Animal Technology (Beijing, China). Mice were housed in barrier facilities under specific pathogen-free conditions with

a 12 hr light/dark cycle at a temperature of 24 ± 2°C and a relative humidity of 50 ± 5%. Mice were fed a standard mouse chow diet, and they consumed food and water ad libitum throughout the experiment. All animal experiments were conducted in accordance with the policies of the Institutional Animal Care Committee of Nankai University (Tianjin, China) and performed under protocol no. 2021-SYDWLL-000029.

## Cell culture

The RAW264.7 mouse macrophage-like cell line (ATCC TIB-71) was obtained from the Shanghai Institute of Biochemistry and Cell Biology of the Chinese Academy of Sciences (Shanghai, China). This cell line originates from ATCC and is not included in the list of commonly misidentified cell lines. Additionally, this cell line is free of mycoplasma contamination. Cells were cultured in RPMI-1640 medium (Gibco #11879020) supplemented with 10% (v/v) fetal bovine serum (FBS, Gibco #10100147) at 37°C with 5% $CO_2$. Cells were seeded in 24-well tissue culture plates at $1 \times 10^5$ cells per well 24 hr before infection.

## Bacterial strains, plasmids, and growth conditions

The bacterial strains and plasmids used in this study are listed in *Supplementary file 1*. The *Salmonella enterica* serovar Typhimurium (STM) strain ATCC 14,028 s was used as the WT strain throughout this study and for the construction of the mutants. Mutant strains were generated using the $\lambda$ Red recombination system with the plasmid pSIM17 (*Jiang et al., 2017*). To construct the complemented strain of Δ*panD*, the amplified DNA fragments of the *panD* ORF and its upstream promoter were digested and inserted into the BamHI–EcoRI site of the low-copy-number plasmid pBR322. To generate the *panD-lux* reporter fusion, the amplification products of the *panD* promoter region were digested and cloned into the XhoI–BamHI site of the plasmid pMS402, which carries a promoter-less *lux*CDABE reporter gene cluster (*Liang et al., 2008*). The sequences of primers used for the construction of the strains are listed in *Supplementary file 2*. All the strains were verified by PCR amplification and sequencing.

Bacterial strains were conventionally grown overnight in Luria–Bertani (LB) medium (10 g/L tryptone, 5 g/L yeast extract, and 10 g/L NaCl) or in N-minimal medium (10 μM $MgCl_2$, 110 μM $KH_2PO_4$, 7.5 mM $(NH_4)_2SO_4$, 0.5 mM $K_2SO_4$, 5 mM KCl, 38 mM glycerol, and 0.1% [w/v] casamino acids) supplemented with appropriate antibiotics at 37 °C with shaking at 180 rpm or on LB agar plates. All antibiotics were used at their standard concentrations (chloramphenicol, 25 μg/mL; kanamycin, 50 μg/mL; ampicillin, 100 μg/mL; gentamicin, 10 or 100 μg/mL) unless otherwise mentioned.

## Growth curve

Bacterial strains were conventionally grown overnight in LB medium. The next day, they were subcultured (1:100) in new LB medium and RPMI-1640 medium or subcultured in N-minimal medium supplemented with glycerol or β-alanine as the sole carbon source at 37 °C with shaking at 180 rpm. To measure the growth of bacteria, 200 μL of the bacterial cultures were transferred to the microplate wells. The absorbance ($OD_{600}$) of the bacteria was measured every half hour for 12 hr with a Spark multimode microplate reader (Tecan).

## Bioluminescent reporter assays

STM WT carrying the *panD-luxCDABE* fusion plasmid was conventionally grown overnight in LB medium, and the next day, the cells were subcultured (1:100) in new LB medium or N-minimal medium for 8 hr. The luminescence of the cultured bacteria (200 μl) was measured with a Spark multimode microplate reader (Tecan). Moreover, the cultured bacteria (100 μl) were serially diluted and plated on LB agar plates to estimate bacterial CFUs. Bacterial CFUs were used to normalize luminescence values.

## *Salmonella* infection of macrophages

Bacterial strains were conventionally grown overnight in LB medium to the late stationary phase, and the next day, the bacteria were diluted to $2 \times 10^6$ CFUs/mL and opsonized in RPMI-1640 medium supplemented with 10% FBS for 15 min. The macrophage monolayers were infected with the opsonized bacteria culture (0.5 mL/well, multiplicity of infection (MOI)=10) and centrifuged at 800×*g*

for 5 min to synchronize infection. The infected cells were incubated for 30 min at 37 °C in 5% $CO_2$ and then washed three times with 1×PBS. Fresh RPMI-1640 medium containing 100 µg/mL gentamicin was added to the infected cells to kill extracellular bacteria. After 1 hr, fresh RPMI-1640 medium containing 10 µg/mL gentamicin was added to the infected cells for the remainder of the experiment. To assess the intracellular growth of *Salmonella*, the infected cells were lysed with 1% Triton X-100 at 2 hpi and 20 hpi, and the abundance of the intracellular bacteria CFUs was estimated on LB agar plates. The relative fold replication of intracellular bacterial strains was denoted as the CFUs recovered at 20 hpi relative to those at 2 hpi. The relative fold change in replication was normalized to the number of RAW264.7 cells. When indicated, 1 mM β-alanine or 100 µM $ZnSO_4$ were added after 1 hr of gentamicin treatment.

## Targeted metabolomics analysis of amino acids in macrophages

Targeted metabolomics was conducted as previously described (*Jiang et al., 2021*), with the combined metabolites of infected cells and intracellular bacteria extracted for analysis. The metabolites have been confirmed to be dominated by the host metabolites (*Jiang et al., 2021*). RAW264.7 cells were mock-infected or infected with STM WT for 8 hr. The cells were harvested and washed with precooled PBS solution to remove the medium. Cellular metabolites were extracted using ice-cold extraction solvent (40:40:20 vol/vol/vol acetonitrile:methanol:water, 0.1 M formic acid), incubated at –20°C for 20 min, and then centrifuged for 10 min at 12,000×g and 4°C to obtain the supernatant. Subsequently, the supernatant was transferred to an LC-MS vial and analyzed using ultrahigh-performance liquid chromatography (Acquity; Waters, Milford, MA, USA) coupled with mass spectrometry (Q Exactive Hybrid Quadrupole-Orbitrap; Thermo Fisher Scientific, Waltham, MA, USA). Metabolites were separated with a Luna $NH_2$ column (2 mm × 100 mm, 3 µm particle size; Phenomenex). Mobile phase A was 20 mM ammonium acetate (pH 9.0), and mobile phase B was acetonitrile containing 0.1% formic acid. The flow rate was 0.4 ml/min. Xcalibur 4.0 software (Thermo Fisher) was used for data acquisition and processing. Metabolite identification was achieved by high-resolution mass and retention time matching to authentic standards. The absolute quantification of amino acids was performed using the standard curve method, and the values were normalized to the cell number. Four biological replicates of each sample were analyzed.

## Immunofluorescence staining

RAW264.7 cells were infected with STM WT or Δ*panD* and the complemented strain c*panD* as described above. After 2 and 20 hr of cultivation, the infected cells were fixed for 15 min in 4% paraformaldehyde, washed in PBS, and permeabilized with 0.1% Triton X-100 in PBS for 15 min. The fixed samples were blocked in 5% bovine serum albumin for 30 min, followed by staining with a FITC-conjugated anti-*Salmonella* antibody (1:100 dilution, Abcam #ab20320) for 1 hr at room temperature in the dark. The nuclei were then stained with DAPI (Invitrogen #D21490) for 2 min. A confocal laser scanning microscope (Zeiss LSM800) and ZEN 2.3 software (blue edition) were used to acquire and analyze the cell images (Objective lens: 40x; The number of intracellular bacteria per infected cell was estimated in random fields by Fiji-ImageJ).

## RNA isolation

RNA was extracted from *Salmonella* strains cultured in N-minimal medium or LB medium. To investigate the expression of the *Salmonella panD* gene inside macrophages, we obtained RNA from intracellular bacteria in RAW264.7 cells at 8 hr post-infection and from bacteria in RPMI medium. RNA was extracted using an EASYspinPlus bacterial RNA rapid extraction kit (Aidlab #RN0802) according to the manufacturer's protocol. RNA quantity and purity were determined using a NanoDrop 2000 spectrophotometer (NanoDrop Technologies). RNA samples were stored at −80 °C before use.

## Quantitative real-time PCR (qRT−PCR)

According to the manufacturer's protocols, qRT−PCR was performed using 2x RealStar Power SYBR qPCR Mix (Genstar #A304) in a QuantStudio 5 real-time PCR system (Applied Biosystems). cDNA was synthesized using a StarScript III RT Kit (Genstar #A232). Each sample was subjected to qRT−PCR in triplicate. The expression level of the 16 S *rRNA* gene was used to normalize that of the target genes. We estimated the expression of each target gene using the $2^{-\Delta\Delta Ct}$ method.

## RNA sequencing and analyses

The STM WT and $\Delta panD$ strains were conventionally grown overnight in LB medium, subcultured (1:100) in N-minimal medium for 8 hr, and then collected by centrifugation for RNA extraction. Sequencing libraries were generated using the NEBNext Ultra RNA Library Prep Kit for Illumina (New England Biolabs) according to the manufacturer's instructions, and sequencing was conducted using the Illumina HiSeq 2000 platform at Shanghai Majorbio Bio-Pharm Technology Co., Ltd. (Shanghai, China). The sequencing data were deposited in the NCBI Sequence Read Archive under accession number (SRA, PRJNA1124637). The clean reads were mapped to the STM ATCC 14028 reference genome (CP001363 and CP001362) by using the short-sequence alignment software Bowtie 2. Gene expression was evaluated using the fragments per kilobase of transcript per million mapped reads (FPKM) method. DEGs in the $panD$ mutant relative to the WT were determined using the R statistical package software EdgeR. The thresholds for statistically significant differences were set to a fold change $\geq 2$ or $\leq 0.5$ and a false discovery rate (FDR) $\leq 0.05$. p-values were adjusted using the Benjamini–Hochberg procedure for controlling the FDR. Enrichment analysis of DEGs was conducted using GO and KEGG enrichment analyses.

## Mouse infection

*Salmonella* strains were conventionally cultured overnight in LB medium, and the next day, they were subcultured (1:100) in new LB medium and grown at 37 °C with shaking at 200 rpm to stationary phase ($OD_{600} \sim 2$). The bacteria were diluted to $5 \times 10^4$ CFUs/mL in 0.9% NaCl. Groups of BALB/c mice were infected i.p. with 0.1 mL of the NaCl suspension. For survival assays, we recorded and monitored the mortality and body weight of the infected mice daily for 10 d. To analyze the bacterial burden of the mouse liver and spleen, we weighed the infected mice first on day 3 post-infection and then harvested the liver and spleen. The liver and spleen of infected mice were homogenized in ice-cold PBS, serially diluted, and plated on LB plates containing the appropriate antibiotics to determine bacterial CFUs. H&E staining of the mouse liver was performed to investigate the histopathological alterations in the liver of infected mice. To evaluate the histopathological alterations in the mouse liver, we harvested the liver of the infected mice on day 5 post-infection. The mouse liver was washed with 0.9% NaCl, fixed with 10% neutral formalin for 48 hr, and subsequently processed for routine paraffin embedding. Paraffin-embedded tissues were sectioned at a thickness of 5 μm and then stained with hematoxylin (Sangon Biotech #E607318) and eosin (Sangon Biotech #E607318) for histopathological examination. The stained sections were then examined by light microscopy (Leica DM2500 LED).

## Flow cytometry

RAW264.7 cells were infected with *Salmonella* WT for 8 hr, in the absence or presence of 1 mM β-alanine, which was added to the infected cells at 1 hr post-infection. The infected cells were fixed for 15 min in 4% paraformaldehyde, washed in PBS, and permeabilized with 0.1% Triton X-100 in PBS for 15 min. The fixed samples were blocked in 5% bovine serum albumin for 30 min, followed by staining with an anti-CD86 antibody (Abcam, #ab119857), an anti-CD163 antibody (Abcam, #ab182422) for 30 min, and a goat anti-rat IgG H&L (Alexa Fluor 488) (Abcam, #ab150165), a donkey anti-rabbit IgG H&L (Alexa Fluor 647) (Abcam, #ab150075) for 30 min in the dark. The infected cells were analyzed using a BD FACSAria Flow Cytometer (BD Biosciences).

## Assessment of intracellular ROS/ RNS levels

RAW264.7 cells were infected with *Salmonella* WT for 8 hr, either in the absence or presence of 1 mM β-alanine. The β-alanine was added to the infected cells at 1 hr post-infection. Following the infection, the cells were washed three times with PBS, and the levels of ROS and RNS were measured using a ROS fluorescence probe (BBoxiProbe O06, Bestbio, #BB-46051) and an RNS fluorescence probe (BBoxiProbe O52, Bestbio, #BB-470567), respectively, in accordance with the manufacturer's protocol. The ROS/RNS levels were normalized to the cell count.

## Assessment of zinc levels in mouse liver and RAW 264.7 macrophages

To assess the zinc level in mouse liver, mice were i.p. infected with approximately 5000 CFUs of *Salmonella* WT or $\Delta panD$ for 3 d, with three mice in each injection group. Following infection, mouse livers were collected and then dissociated into single cells using the gentleMACS/Mouse Liver Dissociation

Kit (Miltenyi Biotec, #130-105-807) in accordance with the manufacturer's protocol. The zinc concentration in mouse liver was determined using a zinc fluorescence probe (Zinquin ethyl ester, MKBio, #MX4516) following the manufacturer's instructions and normalized by the cell count.

To assess the zinc level in RAW 264.7 macrophages, RAW 264.7 cells were infected with *Salmonella* WT or ΔpanD for 8 hr. The infected cells were washed three times with PBS, and zinc concentration was measured using a zinc fluorescence probe (Zinquin ethyl ester, MKBio, MX4516) following the manufacturer's protocol. The zinc concentration was normalized by the cell count.

## Statistical analysis

The data are presented as the mean ± SD. All in vitro experiments were conducted in duplicate and repeated at least three times (n≥3). Mouse assays were performed twice, with at least two mice (n≥2) in each injection group, and the combined data from the two experiments was used for statistical analysis. Statistical analyses were performed using GraphPad InStat software (version GraphPad Prism 9.5.1, San Diego, CA, USA) with two-sided Student's *t*-tests, one-way ANOVA, two-way ANOVA, log-rank Mantel–Cox tests, or Mann–Whitney U tests according to the test requirements (as stated in the figure legends). A *p*-value <0.05 indicated a statistically significant difference. ns represents no statistical significance.

## Acknowledgements

This work was supported by the National Natural Science Foundation of China (NSFC) Program (grant no. 32170110, 32270191), the Natural Science Foundation of Tianjin (grant no. 22JCYBJC01060), and the Fundamental Research Funds for the Central Universities, Nankai University (grant no. 63241588, 63243161).

## Additional information

### Funding

| Funder | Grant reference number | Author |
|---|---|---|
| National Natural Science Foundation of China | 32170110 | Lingyan Jiang |
| Natural Science Foundation of Tianjin Municipality | grant no. 22JCYBJC01060 | Lingyan Jiang |
| Fundamental Research Funds for the Central Universities | 63241588 | Lingyan Jiang |
| National Natural Science Foundation of China | 32270191 | Lingyan Jiang |
| Fundamental Research Funds for the Central Universities | 63243161 | Lingyan Jiang |

The funders had no role in study design, data collection and interpretation, or the decision to submit the work for publication.

### Author contributions

Shuai Ma, Data curation, Formal analysis, Validation, Visualization, Writing – original draft, Writing – review and editing; Bin Yang, Data curation, Software, Formal analysis; Yuyang Sun, Xinyue Wang, Houliang Guo, Ruiying Liu, Ting Ye, Chenbo Kang, Jingnan Chen, Validation; Lingyan Jiang, Funding acquisition, Writing – original draft, Project administration, Writing – review and editing

### Author ORCIDs

Shuai Ma http://orcid.org/0009-0006-8348-0805
Lingyan Jiang https://orcid.org/0000-0001-8580-606X

## Ethics

All animal experiments were conducted in accordance with the policies of the Institutional Animal Care Committee of Nankai University (Tianjin, China) and performed under protocol no. 2021-SYDWLL-000029.

Reviewer #1 (Public review): https://doi.org/10.7554/eLife.103714.4.sa1
Reviewer #3 (Public review): https://doi.org/10.7554/eLife.103714.4.sa2
Author response https://doi.org/10.7554/eLife.103714.4.sa3

---

## Additional files

### Supplementary files

Supplementary file 1. Strains and plasmids used in this study.

Supplementary file 2. Primers involved in this study.

MDAR checklist

### Data availability

The RNA-seq data generated in this study have been deposited in the NCBI Sequence Read Archive (SRA) database under the accession number PRJNA1124637. All other data associated with this study are available in the main text and supplementary materials. Source data are provided in supplementary data.

The following dataset was generated:

| Author(s) | Year | Dataset title | Dataset URL | Database and Identifier |
|---|---|---|---|---|
| Shuai M | 2024 | A Study on the Pathogenic Mechanism of *Salmonella* | https://www.ncbi.nlm.nih.gov/bioproject/PRJNA1124637 | NCBI BioProject, PRJNA1124637 |

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
