## [Editor Report · eLife Assessment]

The authors use a multidisciplinary approach to provide a link between Beta-alanine and S. Typhimurium (STM) infection and virulence. This **valuable** work shows how Beta-alanine synthesis mediates zinc homeostasis regulation, possibly contributing to virulence. The work is **convincing** as it adds to the existing knowledge of metabolic flexibility displayed by STM during infection.

---

## [Referee Report · Reviewer #1 (Public review)]

Summary:

Ma & Yang et al. report a new investigation aimed at elucidating one of the key nutrients S. Typhimurium (STM) utilizes with the nutrient-poor intracellular niche within macrophages, focusing on the amino acid beta-alanine. From these data, the authors report that beta-alanine plays important roles in mediating STM infection and virulence. The authors employ a multidisciplinary approach that includes some mouse studies, and ultimately propose a mechanism by which panD, involved in B-Ala synthesis, mediates regulation of zinc homeostasis in *Salmonella*.

Strengths and weaknesses:

The results and model are adequately supported by the authors' data. Further work will need to be performed to learn whether the Zn2+ functions as proposed in their mechanism. By performing a small set of confirmatory experiments in S. Typhi, the authors provide some evidence of relevance to human infections.

Impact:

This work adds to the body of literature on the metabolic flexibility of *Salmonella* during infection that enable pathogenesis.

---

## [Referee Report · Reviewer #3 (Public review)]

*Salmonella* is interesting due to its life within a compact compartment, which we call SCV or *Salmonella* containing vacuole in the field of *Salmonella*. SCV is a tight-fitting vacuole where the acquisition of nutrients is a key factor by *Salmonella*. The authors among many nutrients, focused on beta-alanine. It is also known that *Salmonella* requires beta-alanine from many other studies. The authors have done in vitro RAW macrophage infection assays and In vivo mouse infection assays to see the life of *Salmonella* in the presence of beta-alanine. They concluded by comprehending that beta-alanine modulates the expression of many genes including zinc transporters which is required for pathogenesis.

[Editors' note: The authors have appropriately addressed the previous reviewers' concerns.]

---

## [Author Response]

The following is the authors’ response to the previous reviews

**Reviewer #1 (Public review):**
Summary:Ma & Yang et al. report a new investigation aimed at elucidating one of the key nutrients S. Typhimurium (STM) utilizes with the nutrient-poor intracellular niche within macrophage, focusing on the amino acid beta-alanine. From these data, the authors report that beta-alanine plays important roles in mediating STM infection and virulence. The authors employ a multidisciplinary approach that includes some mouse studies, and ultimately propose a mechanism by which panD, involved in B-Ala synthesis, mediates regulation of zinc homeostatisis in *Salmonella*.Strengths and weaknesses:The results and model are adequately supported by the authors' data. Further work will need to be performed to learn whether the Zn2+ functions as proposed in their mechanism. By performing a small set of confirmatory experiments in S. Typhi, the authors provide some evidence of relevance to human infections.Impact:This work adds to the body of literature on the metabolic flexibility of *Salmonella* during infection that enable pathogenesis.
**Reviewer #1 (Recommendations for the authors):**
No further suggestions. The authors have adequately addressed my prior concerns through new data and revisions to the text.

Thank you for considering this work. We appreciate your efforts in aiding us to improve our manuscript.

**Reviewer #3 (Public review):**
Summary:*Salmonella* is interesting due to its life within a compact compartment, which we call SCV or *Salmonella* containing vacuole in the field of *Salmonella*. SCV is a tight-fitting vacuole where the acquisition of nutrients is a key factor by *Salmonella*. The authors among many nutrients, focussed on beta-alanine. It is also known that *Salmonella* requires beta-alanine from many other studies. The authors have done in vitro RAW macrophage infection assays and In vivo mouse infection assays to see the life of *Salmonella* in the presence of beta-alanine. They concluded by comprehending that beta-alanine modulates the expression of many genes including zinc transporters which is required for pathogenesis.Strengths:Made a couple of knockouts in *Salmonella* and did transcriptomic to understand the global gene expression patternWeaknesses:(1) Transport of Beta-alanine to SCV is not yet elucidated. Is it possible to determine whether the Zn transporter is involved in B-alanine transport?

Thank you for the comment. Following your suggestion, we investigated the growth of *Salmonella* WT and the ∆*znuA* mutant cultured in N-minimal and M9 minimal medium, with β-alanine as the sole carbon source. We observed no significant difference in growth kinetics between the ∆*znuA* mutant and WT strain under either culture condition (please refer to Author response image 1). The results indicate that ZnuA is not involved in β-alanine transport in *Salmonella*.

(2) Beta-alanine can also be shuttled to form carnosine along with histidine. If beta-alanine is channelled to make more carnosine, then the virulence phenotypes may be very different.

Our study reveals that β-alanine availability, whether obtained from the host or synthesized de novo via the *panD*-dependent pathway, is important for *Salmonella* pathogenesis. We have shown that β-alanine influences *Salmonella* intracellular replication and in vivo virulence partly by enhancing the expression of the zinc transporter genes.

Although β-alanine can also be shuttled to form carnosine along with histidine in animals, the *Salmonella* genome lacks canonical carnosine synthase (CARNS) orthologs that catalyze the condensation of β-alanine and histidine into carnosine. Therefore, we believe that the carnosine biosynthetic pathway does not influence the virulence phenotypes of *Salmonella*.

(3) Some amino acid transporters can be knocked out to see if beta-alanine uptake is perturbed. Like ArgT transport Arginine, and its mutation perturbs the uptake of beta-alanine. What is the beta-alanine concentration in the SCV? SCVS can be purified at different time points, and the Beta-alanine concentration can be measured

Thank you for the comment. As suggested, we have investigated the role of other amino acid transporters in the uptake of β-alanine. In *E. coli*, GabP transports γ-aminobutyric acid (GABA), a structural analogue of β-alanine, and may also transport β-alanine (J Bacteriol. 2021, 203(4):e00642-20). Nevertheless, *Salmonella* ∆*gabP* mutant displayed no growth defect in minimal medium with β-alanine as the sole carbon source (Figure 1_figure Supplement 7, Figure 1_figure Supplement 8), indicating that GabP is not involved in β-alanine uptake in *Salmonella*. Strikingly, the Δ*argT* mutant—defective in arginine uptake—showed markedly decreased growth in the minimal medium with β-alanine as the sole carbon source (Figure 1F)，suggesting that ArgT also transports β-alanine in *Salmonella*. We have added the results in the revised manuscript (lines 167-179).

It has been reported that ArgT is essential for *Salmonella* replication within macrophages and full virulence in vivo (PloS one. 2010, 5(12):e15466). Given that ArgT is involved in both arginine and β-alanine uptake (as verified in this study), whether the attenuated virulence of the ∆*argT* mutant is due to a deficiency in β-alanine or arginine requires further investigation. We have also included a discussion on this issue (lines 409-415).

In this work, to avoid delays and alterations in metabolite concentrations during the isolation of bacterial contents from macrophages, we directly assessed the combined metabolite concentrations within infected cells and *Salmonella*. It has been previously verified that these metabolites are primarily of host origin (Nat Commun. 2021, 12(1):879.). We noted a decrease in β-alanine levels in macrophages infected with *Salmonella*. The process of separating SCV is intricate and encompasses dissociation and sonication (Nat Commun. 2018, 9(1):2091). These steps may potentially result in alterations of metabolite concentrations during the separation procedure. Therefore, we did not measure the β-alanine concentration in the SCV.

**Reviewer #3 (Recommendations for the authors):**
The Authors have done meticulous experiments to address the questions asked by the reviewers. My one question of beta-alanine transport inside the SCV remains undone, though the authors have tried.Was Zinc transporter mutant checked? It is possible that the Zn transporter can take up Beta-alanine.

Thank you for the comment. Following your suggestion, we investigated the growth of *Salmonella* WT and the ∆*znuA* mutant cultured in N-minimal and M9 minimal medium, with β-alanine as the sole carbon source. We observed no significant difference in growth kinetics between the ∆*znuA* mutant and WT strain under either culture condition (please refer to Author response image 1). The results indicate that ZnuA is not involved in β-alanine transport in *Salmonella*.

Additionally, we have investigated the role of other amino acid transporters in the uptake of β-alanine and have ultimately identified that ArgT, the arginine transporter, is involved in the uptake of β-alanine in *Salmonella* (please refer to our previous response).